# Pancreatic Transdifferentiation Using β-Cell Transcription Factors for Type 1 Diabetes Treatment

**DOI:** 10.3390/cells11142145

**Published:** 2022-07-08

**Authors:** Alexandra L. G. Mahoney, Najah T. Nassif, Bronwyn A. O’Brien, Ann M. Simpson

**Affiliations:** School of Life Sciences, University of Technology Sydney, Sydney 2007, Australia; alexandra.l.mahoney@student.uts.edu.au (A.L.G.M.); najah.nassif@uts.edu.au (N.T.N.); bronwyn.obrien@uts.edu.au (B.A.O.)

**Keywords:** type 1 diabetes, gene therapy, viral vectors, beta-cell transcription factors, pancreatic transdifferentiation

## Abstract

Type 1 diabetes is a chronic illness in which the native beta (β)-cell population responsible for insulin release has been the subject of autoimmune destruction. This condition requires patients to frequently measure their blood glucose concentration and administer multiple daily exogenous insulin injections accordingly. Current treatments fail to effectively treat the disease without significant side effects, and this has led to the exploration of different approaches for its treatment. Gene therapy and the use of viral vectors has been explored extensively and has been successful in treating a range of diseases. The use of viral vectors to deliver β-cell transcription factors has been researched in the context of type 1 diabetes to induce the pancreatic transdifferentiation of cells to replace the β-cell population destroyed in patients. Studies have used various combinations of pancreatic and β-cell transcription factors in order to induce pancreatic transdifferentiation and have achieved varying levels of success. This review will outline why pancreatic transcription factors have been utilised and how their application can allow the development of insulin-producing cells from non β-cells and potentially act as a cure for type 1 diabetes.

## 1. Introduction

Diabetes mellitus has been recognised by the United Nations as a debilitating, chronic and costly disease with an increasing prevalence due to an ageing population [1]. There are approximately 422 million people worldwide living with diabetes, with this number steadily increasing every year [2]. Type 2 diabetes (T2D) results from the resistance of peripheral tissues (notably liver and muscle) to the actions of insulin due to beta (β)-cell dysfunction and/or increased insulin demands due to obesity. Gestational diabetes is diagnosed during pregnancy and presents as hyperglycaemia due to glucose intolerance [3,4]. Type 1 diabetes (T1D) results from the T-cell-mediated destruction of insulin-producing β-cells in the pancreas and is the focus of this review [5].

T1D is commonly characterised by the loss of insulin function, due to the autoimmune elimination of β cells resulting in chronic hyperglycaemia. Currently, the most common treatment regime for T1D is the multiple daily delivery of exogenous insulin [6]. In many cases however, this treatment is not well managed by patients and the injection of bolus doses of insulin cannot mimic the minute-to-minute responsiveness of the pancreatic β-cell. Consequently, patients experience episodes of hypo- and hyperglycaemia, which significantly increase morbidity and mortality. Hyperglycaemia leads to adverse chronic complications, including retinopathy, kidney failure and neuropathy. Hypoglycaemia potentially causes seizures, coma, or death [6,7].

The insulin pump, also known as the artificial pancreas, is an alternative to insulin injections for T1D, and some T2D patients [8]. The device measures blood glucose concentrations and delivers insulin accordingly in an effort to improve the accuracy of the insulin dosage. Although the insulin pump has proven to be more effective than daily injections, the device shows a time lag between the increase in blood glucose levels and the release of insulin. Additionally, there can be discrepancies between the blood glucose values measured by the device and the actual levels [8].

The only cures currently available for T1D are whole pancreas and islet transplantation, both of which are unrealistic options due to the limited amount of donor tissue and the need for lifelong immunosuppression to combat recurrent autoimmunity [9]. Given that the number of people diagnosed with T1D has been increasing significantly during the last decade, coupled with the issues related to treatments currently available, an alternative therapeutic approach is urgently required [1].

There has been extensive research into alternative approaches aimed to accurately and efficiently deliver insulin while negating the need for both donor tissue and immunosuppressive drugs. These approaches include different strategies to generate ‘artificial’ β-cells to allow for the physiological release of insulin [10]. The transdifferentiation of non-pancreatic cells has been extensively researched as a method to produce insulin-secreting cells. The delivery of the insulin gene to susceptible cell types using a viral vector can result in the transdifferentiation of the target cell so that it ultimately produces, stores and secretes insulin in a glucose responsive manner, akin to that of a pancreatic β-cell [11]. The transdifferentiation of a patient’s own cells would avoid the destructive allogenic responses, and limitations of donor tissue availability. Additionally, as these cells would not be pancreatic β-cells per se, they may be resistant to recurrent anti-islet immune responses.

This review will discuss current research areas for therapeutic approaches for the treatment of T1D, and it will specifically compare the different ways in which β-cell transdifferentiation of non-insulin-producing cells through gene therapy can be initiated and maintained.

## 2. Pancreatic Development and Function

Beta-cells are located in the islets of Langerhans in the pancreas, and they play the major role in the maintenance of normoglycaemia. This is due to the ability of β-cells to store produced insulin and then secrete it in response to minor changes in blood glucose concentrations [12]. Knowledge of how β-cell development occurs from the embryonic stage through to the mature stage has allowed a greater understanding of the elements required to generate a functional ‘artificial’ β-cell. Significant research has focused on gaining an understanding of the mechanisms of development and the function of β-cells, including the regulated expression of critical β-cell transcription factors and the need for a glucose sensing system.

During embryonic pancreatic development, transcription factors play a significant role in islet cell differentiation and phenotype maintenance during adult life by regulating the expression of pancreatic hormones (Figure 1). Forkhead box factor *FoxA2* is present in the endodermal layer from which the pancreas develops. Deletion of *FoxA2* disrupts the formation of the endoderm and consequently stunts the development of the pancreas and islet cells [13]. *FoxA2* is also involved in further development of the pancreas through its effect on the expression of other key transcription factors including pancreas/duodenum homeobox protein 1, *Pdx1* [14]. *Pdx1* is present in both the endoderm and pancreatic buds and is responsible for early pancreatic development. Deletion of this transcription factor in mice results in the complete absence of the pancreas [15].

During development of the pancreas, differentiation of pancreatic cells to either endocrine or non-endocrine cells is controlled by helix–loop–helix factors: *Ngn3* (Neuro-genin 3), *Hes1* (hairy and enhancer of split 1) and *NeuroD1* (neurogenic differentiation 1) [16,17]. *Ngn3* regulates the production of endocrine precursor cells and endocrine cell types, and deletion of Ngn3 results in the absence of endocrine cells [18]. *NeuroD1* regulates the growth and proliferation of these endocrine cells [17]. *Hes1* directs the differentiation to non-endocrine cell lineages including exocrine cells and duct cells. In the development stage, *Ngn3* and *Hes1* interact through notch signalling [19], an important pathway in driving endocrine cell formation and differentiation because it negatively regulates *Ngn3* expression [20]. High notch signalling results in increased activity of *Hes1*, which blocks *Ngn3* and facilitates non-endocrine differentiation [21,22]. On the other hand, low notch signalling enables Sox9^+^ expression, which increases *Ngn3* expression levels and consequently stimulates endocrine differentiation [23]. Through these interactions, the expression and timing of the different β-cell transcription factors allow for the formation of insulin-producing β-cells during the early development of the pancreatic cells.

In the developed pancreas, transcription factors allow pancreatic β-cells to regulate the transcription, translation, storage, and eventual secretion of insulin. Upon a change in the rate of glucose metabolism, transcription factors promote expression of the insulin gene. Transcription factors, such as *Pdx1*, V-maf musculoaponeurotic fibrosarcoma oncogene homolog A (*MafA)* and insulin enhancer binding protein 1 (*Isl1*) work at both the transcriptional and translational level. *Pdx-1*, *MafA* and *Isl1* play an important role in the mature pancreas by activating insulin production [24,25,26]. These transcription factors work in conjunction with the glucose sensing system to allow glucose stimulated insulin release.

The primary function of β-cells is to release appropriate amounts of insulin rapidly in response to minute changes in blood glucose concentrations [27]. The β-cells are able to do this through a glucose sensing system which involves mechanisms that directly recognise changes in physiological glucose concentration across a broad range (5–20 mM) [28]. In addition to β-cells, a glucose sensing system is present in a range of cell types including liver and intestinal cells [27]. The system is composed of the glucose transporter-2 (GLUT2) and glucokinase, with the presence of both components being necessary for glucose stimulated insulin secretion to occur. GLUT2 facilitates the transport of glucose across plasma membranes and allows appropriate equilibration of extracellular and intracellular glucose [29]. It is also the major glucose transporter in liver cells in humans, up taking glucose in the fed state and releasing it when glucose levels fall. Glucokinase controls the rate at which glucose-stimulated insulin secretion occurs through enzymatically phosphorylating glucose to produce glucose-6-phosphate [30]. This sensing system allows insulin secretion to be extremely responsive and precise as it can sense small changes in physiological glucose concentrations and then adjust insulin release by β-cells accordingly.

## 3. Transdifferentiation to β-Cells

There has been significant research into the various ways in which artificial β-cells can be developed to replace the insulin-producing cells that are lost in T1D patients. Previously, it was thought that T1D could be treated by the regeneration of β-cells after an insult to the pancreas, for example, through a partial pancreatectomy. Although this was successful in mice, in humanized models, it was ineffective in stimulating sufficient β-cell regeneration [31,32]. Researchers have now considered producing β-like cells from other cell types, and one such way to do this is through a process that has been termed transdifferentiation. Transdifferentiation is the conversion of a cell from a particular lineage to one of another lineage. Although there is still more to understand regarding the transdifferentiation process, there has been some key criteria identified as a way in which to identify whether transdifferentiation has occurred. These criteria include proving the cell did not undergo division before entering the differentiated state and showing they possess a changed phenotype that is mature and stable [33]. This method of developing a specific cell type has been researched significantly to treat different conditions, including neurodegenerative and corneal diseases, and is now being explored in the context of T1D [34,35]. This is commonly accomplished through gene delivery strategies to induce the expression of specific transcription factors, which facilitate transdifferentiation to the desired cell lineage. Exposing target cells to specific transcription factors that are involved in the development of another cell type, aims to mimic the conditions during embryonic development that cause the differentiation of cells along a particular lineage. For T1D treatment, transdifferentiated cells must be able to produce, store, and secrete insulin in response to glucose order to be a functional β-cell replacement. Currently, there is significant research into initiating transdifferentiation of both stem cells and somatic cells through either in vitro or in vivo exposure to transcription factors.

### 3.1. Differentiation of Stem Cells

Stem cells have been investigated as potential replacements for pancreatic insulin-producing cells due to their plasticity and favourable immunomodulatory properties. These cells can possess anti-inflammatory properties and mediate immunoregulatory effects, which is useful in the treatment of diseases which are caused by excessive pro-inflammation [36]. There are a number of different stem cells that have been investigated, including embryonic, mesenchymal, and induced pluripotent stem cells. Embryonic stem cells (ESCs) harvested from the blastocyst stage are favourable candidates as they are able to differentiate along any cell lineage, including insulin-producing cells [37]. The use of ESC-derived pancreatic endoderm cells for T1D treatment has been investigated and is currently undergoing clinical trials. An encapsulation device is used to implant these cells into T1D patients and has proven to allow for C-peptide release but lacks in stable, long-term insulin release [38,39]. Although a promising treatment, a viable embryo needs to be created and destroyed in order to harvest ESCs, which present significant ethical and donor tissue availability issues.

Alternatively, mesenchymal stem cells present an opportunity for an autologous transplant from the donor’s umbilical cord blood, adipose tissue, amniotic fluid, or bone marrow [40]. Mesenchymal stem cells obtained from murine bone marrow have been shown to transdifferentiate into insulin-producing cells that respond to changes in glucose concentrations in vitro and in streptozotocin (STZ) diabetic immunodeficient mice [41]. However, there have been conflicting results as to whether adult bone marrow-derived stem cells are capable of transdifferentiation to β-cells in vivo [42]. Although the use of mesenchymal stem cells for pancreatic transdifferentiation has been limited to date, they have recently been explored for their immunomodulatory properties to improve post-transplantation maintenance of insulin-producing cells [43]. Numerous investigations of the co-transplantation of mesenchymal stem cells with islet cells have been reported; however, this is beyond the scope of this review [44,45].

Alternatively, developing induced pluripotent stem cells (iPSCs) by dedifferentiating somatic cells back to an embryonic stem cell-like state is another way in which insulin-producing cells may be generated. Once obtained, the cells can be transdifferentiated to become insulin-producing cells using pancreatic transcription factors [46]. The benefit of using iPSCs is that they can be easily harvested from any adult somatic cells, and this method has been shown to be successful using both human and mouse fibroblasts [47,48]. Another favourable aspect of iPSCs is that the somatic cells to be used for dedifferentiation can be taken directly from the patient, allowing autologous transplantation and thereby limiting the probability of rejection [49]. Having confirmed that these iPSCs resemble embryonic stem cells, combined with a reduction in the level of ethical issues related to their use, induced pluripotent stem cells are an attractive option for a cure for T1D [50].

Despite these favourable characteristics, the use of stem cells as a source for the development of insulin-producing cells still has a number of difficulties to overcome. There is still conflicting research on how the immune system responds to these cells post transplant. It is possible that even after the cells have undergone transdifferentiation, these cells will then be susceptible to the same autoimmune attack that destroyed the original β-cell population. Furthermore, the tumorigenic nature of some transdifferentiated stem cells is another reason why this therapy may not be the solution for treating T1D [51].

### 3.2. Transdifferentiation of Somatic Cells

The benefit of attempting transdifferentiation from non β-cell precursors is that it may minimize the reoccurrence of anti-β-cell immune responses as the cells may not possess the complete set of autoantigens inherent to pancreatic β-cells. Studies in our laboratory have shown that insulin-secreting cell lines are resistant to the detrimental effects of classical β-cell cytotoxins and pro-inflammatory cytokines that play a principal role in the autoimmune process of diabetes [52,53]. Additionally, no infiltrates of immune cells were found in a non-obese diabetic (NOD) mouse engineered to express insulin in their livers [54]. Another study by Lipes et al. [55] indicated that the same is true for insulin-secreting pituitary cells, which showed no autoimmune destruction.

Different somatic cells including pancreatic, liver, intestinal, fibroblasts, skeletal myocytes, and muscle cells have been investigated to assess their abilities to transdifferentiate to β-cells due to shared pancreatic β-cell characteristics. Additionally, some of these cell types derive from the same endodermal region. For example, pancreatic and liver cells have common endoderm precursors, and have been widely explored as candidates for β-cell transdifferentiation.

Different endocrine and exocrine pancreatic cells including α-cells, duct epithelial cells, acinar cells, and pancreatic progenitor cells have been investigated for their potential use as precursors for the production of insulin-producing cells. The use of other pancreatic cells to develop β-cells is a favourable option for β-cell transdifferentiation as they are developmentally similar [56]. Specifically, α-cells have been researched intensively for their ability to undergo β-cell transdifferentiation as they are located within islets, which would be beneficial for their functionality after differentiation [57,58].

Of all the candidates for β-cell transdifferentiation, liver cells are perhaps the cell type of choice due to some shared key characteristics. Liver cells develop from the same endodermal region as the pancreas, which allows the transdifferentiation to pancreatic cells to be more feasible compared to some other cell types. Liver cells also possess the glucose sensing system, which is necessary for rapid and precise detection of changes in physiological glucose levels [27]. Although they neither possess secretory granules nor pro-insulin processing enzymes, the delivery of furin-cleavable insulin (INS-FUR), which has a furin cleavage site allowing the cleavage of proinsulin to insulin and c-peptide, overcomes this shortcoming and allows insulin to be processed in the liver cells. Liver cells have also proved to be able to maintain their function while also being able to store and secrete insulin. In a study by Ren et al. [59], liver cells of NOD mice were engineered to secrete and store insulin and exhibited normal glucose tolerance. At no time during the study did the mice exhibit raised liver enzymes or any evidence of autoimmune attack, indicating that normal liver function was maintained, and the livers were not subject to autoimmune destruction. Similarly, in chemically induced diabetic rats, differentiation of a portion of the liver to secrete insulin had no effect on liver enzymes, which remained normal throughout the extensive study in which the animals were maintained for 500 days and remained normoglycemic for the period [60].

Accordingly, liver cells have been extensively explored as precursors for β-cell regeneration, in studies conducted in vitro, ex vivo and in vivo [59,61,62]. Melligen cells, which were developed from a liver cell line that expresses endogenous β-cell transcription factors, were able to store insulin and reverse T1D in immunodeficient NOD mice after the subcutaneous injection of the cells [61]. Normoglycaemia was maintained, and once the graft was removed, the mice reverted to hyperglycaemia. Further, Huh7 cells, transduced with the INS-FUR gene are also an example which shows the ability of a liver cell to transdifferentiate into a glucose responsive, insulin-producing cell [63]. Primary hepatocytes have also been transdifferentiated using β-cell transcription factors, both in vitro and ex vivo, with promising results in murine models [64,65]. Using an adenoviral vector to deliver a range of β-cell transcription factors, hepatocytes were able to begin to produce insulin in culture and when transplanted back into diabetic mice the animals became normoglycaemia. Hepatocytes have also been the target of non-viral insulin delivery using a minicircle DNA-based system which resulted in hepatic insulin secretion and correction of diabetic hyperglycaemia [66].

Therefore, given their shared characteristics with β-cells and their ability to partially transdifferentiate and produce insulin, liver cells are a favourable target for β-cell replacement therapy.

## 4. Gene Delivery Strategies

In order to deliver genes to a specific cell type for therapeutic purposes, an effective and clinically applicable vector is required. Currently used gene delivery tools are generally either non-viral delivery or viral vectors. An ideal vector is one that would be able to safely deliver genes to a certain cell type, without initiating an immune response, and permanently cure a disease. Despite the benefits of low immunogenicity associated with non-viral vectors, they have a low transfection rate, and consequently they only provide transient gene expression. As non-viral vectors do not integrate into the host chromosome, they require multiple injections, and in many cases, permanent cure of the disease is not possible [67]. Conversely, certain viral vectors are more likely to illicit an immune response, but their expression and transduction efficiency have the potential to be a long-term solution for T1D. Viral vectors most commonly used in T1D research include retroviral, adeno and adeno-associated, and lentiviral vectors (Table 1).

### 4.1. Retroviral Vectors

Retroviral vectors have been developed from a disabled murine virus and are frequently used in gene therapy studies [81]. Using a retroviral vector is beneficial as it integrates into the host’s genome, allowing for sustained gene expression. However, this introduces the risk of insertional mutagenesis, potentially leading to tumour development. For example, two patients who had received retroviral therapy to cure severe combined immunodeficiency (SCID)-X1 disease developed leukaemia three years after the treatment, raising biosafety concerns [82]. Furthermore, retroviruses can only integrate into dividing cells, thereby limiting their broad applicability. A vector with less associated risk and a large cell target range would be more suitable for T1D treatment [83].

### 4.2. Adenoviral Vectors

Adenoviruses are non-enveloped, double stranded DNA viruses, which are members of the Adenoviridae family [84]. The original virus is the causative agent of conjunctival and respiratory disease, and at least 47 human adenovirus types have been identified to date [85]. Unlike retroviral vectors, adenoviral vectors (AV) can transfer genes to both dividing and non-dividing cells, and they have a large cassette capacity of 8 kb. Additionally, they can be produced in high titres and can deliver genes at a high multiplicity of infection [81,86]. These characteristics have made AV one of the most widely used viral vector systems for both in vitro studies and clinical trials. However, AV does not integrate into the chromosome of the host, and therefore, the expression of target genes can be transient. They also can elicit a strong immune response, and immune suppression may be required, thereby limiting their clinical applicability [87]. Accordingly, the aforementioned vector systems are not the optimal choice for production of artificial β-cells.

### 4.3. Adeno-Associated Viral Vectors

Adeno-associated viruses are small virions containing ssDNA molecules. These viruses are members of the *Dependovirus* genus as they require co-infection with other viruses, such as adenoviruses, which can transduce both dividing and non-dividing cells with long-term gene expression [88]. Adeno-associated viruses express the rep, cap, and aap viral genes, but when developing the vector, these viral proteins are removed in order to improve the safety profile [89]. The site-specific nature of their expression further increases their safety as it limits potential oncogenic consequences. However, these vectors have limited gene cargo capacity (4.8 kB), and large numbers of humans have pre-existing antibodies against AAV variants, which negatively impacts gene transfer [90]. Previous studies have used AAV vectors in order to treat T1D, one study using the vector to deliver insulin and transcription factors was unsuccessful in inducing pancreatic transdifferentiation and consequently may not be an appropriate vector to use for T1D research [62].

### 4.4. Lentiviral Vectors

Lentiviral vectors are a type of retrovirus, and, like retroviral vectors, they incorporate into the host genome resulting in long term gene expression. They are able to transduce both dividing and non-dividing cells making them an ideal choice for a range of gene delivery applications. The insertion into the host genome shows less preference for integration in close proximity to proto-oncogenes, thereby limiting the risk of insertional mutagenesis [91]. Additionally, lentiviral vectors do not elicit a strong immune response and are therefore ideal for clinical application. Most lentiviral vectors have been developed from the human immunodeficiency virus (HIV), which has led to some biosafety concerns. Using a third-generation, self-inactivating lentiviral vector instead of the second-generation vectors significantly reduces the biosafety risk of viral replication and development of HIV because the long terminal repeat promotor has been removed from these vectors [92].

## 5. Delivery of β-Cell Transcription Factors to Induce Transdifferentiation

Beta-cell transcription factors have been used in combination with viral vectors to induce pancreatic transdifferentiation of non-insulin-producing cells (Table 1). While their use has witnessed both successes and setbacks, they offer an overall promising solution when compared to the shortcomings of treatments that are currently available.

Using an adenoviral vector in transgenic mice, Matsouka et al. [93] investigated the expression of *Pdx1* and *MafA* in α-cells. The expression of either *Pdx1* or *MafA* alone was unable to induce the production of insulin in α-cells, and they continued to secrete glucagon as normal. This finding has been corroborated by a number of studies in which overexpression of a single transcription factor has proved to be insufficient to reprogram non β-cells to provide permanent diabetes reversal [94,95]. Although a number of studies have reported successful transdifferentiation of cells using *Pdx1* alone, the reproducibility of these results is limited and has only proved to be successful in a small number of studies. Additionally, the delivery of only *Pdx1* can stimulate transdifferentiation along a pancreatic exocrine lineage, thereby inducing tissue destruction. This was demonstrated after the in vivo delivery of *Pdx1* to the liver. Consequently, it is necessary to elucidate the combination of β-cell transcription factors that will allow for successful and complete transdifferentiation.

Delivery of a combination of transcription factors has witnessed varying success across a range of cell types. Co-expression of *Pdx1* and *MafA* in the study conducted by Matsouka et al. [93] exhibited the production of insulin after two weeks of transduction of the α-cells with transcription factors with no glucagon detected. There was a similar finding in a study conducted by Xiao et al. [96] in which *Pdx1* and *MafA* were delivered via AAV to the pancreatic duct in NOD mice. This was successful when transdifferentiation occurred in vitro, and normoglycaemia was achieved in vivo within 4 weeks of viral infusion to the pancreatic duct and continued for up to 4 months after delivery as β-cells were able to develop from an α-cell origin. However, hyperglycaemia re-occurred, suggesting that *Pdx1* and *MafA* are key transcription factors in efforts to reverse diabetes; however, these transcription factors were not able to achieve complete pancreatic transdifferentiation when delivered on their own.

Another common transcription factor combination is the co-delivery of *Pdx1*, *Ngn3* and *MafA* [74,76,77,78]. This combination has shown success in initiating pancreatic transdifferentiation through viral delivery, especially in the liver. Delivering *Pdx1*, *NeuroD1* and *MafA* to liver cells was shown to cause increased insulin production capacity; however, the glucose responsiveness of these cells was unable to match that of native β-cells [97]. Another study has investigated the co-delivery of transcription factor *Isl1* with *Pdx1*, *Ngn3* and *MafA* to determine whether this would allow for a more stable and long-term transdifferentiation of cells [72]. This study found that hyperglycaemia was ameliorated in diabetic mice for 28 days with enhanced glucose responsiveness and prolonged insulin production. Although promising results have been observed thus far, complete and sustained β-cell transdifferentiation and glucose responsiveness has yet to be achieved in vivo.

Considering these findings, the delivery of transcription factors is complex and multifaceted. There is still insufficient evidence to ascertain the most effective combination of transcription factors to cause pancreatic transdifferentiation. Similarly, there has been insufficient and inconclusive research to provide the optimal delivery route and combination of transcription factors to deliver to a certain cell type. However, the primary findings reported here indicate that the use of a viral vector to deliver transcription factors to cells of liver or pancreatic origins shows promise as a strategy to reverse T1D. Success in this area of research would mean T1D patients may no longer require donor tissue or immune suppressant drugs to cure their disease, being a remarkable and highly impactful finding.

## 6. Conclusions and Future Directions

Pancreatic transdifferentiation has shown a lot of promise as a way in which T1D can potentially be cured. Currently, there is extensive research dedicated to triggering pancreatic transdifferentiation of stem cells or somatic cells by the expression of β-cell transcription factors. This method has shown success and is proving to be a positive path for research in the future. Other methods of inducing pancreatic transdifferentiation have also been explored, including triggering spontaneous pancreatic transdifferentiation through the expression of insulin or using glucose responsive promotors in a range of cells. In conclusion, the development of insulin-producing cells from non β-cell lineages could alleviate the issues seen with the currently available T1D treatments and allow those with the condition an improved quality of life.

## Figures and Tables

**Figure 1 cells-11-02145-f001:**
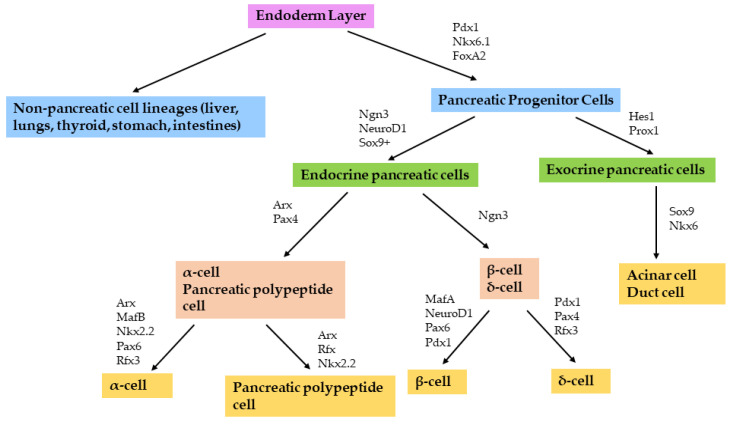
Representation of the hierarchy of the human pancreatic transcription factors involved in the development and differentiation of pancreatic cells. FoxA2 is present in the pancreatic buds, which develop from the endodermal layer. Pdx1 allows the development of the pancreas and differentiation of pancreatic cells through interaction with other transcription factors: Hes1 and Prox1 to non-endocrine cell lineages and Ngn3, NeuroD1 and Sox9^+^ to endocrine cells. Pax4 and MafA then are thought to mediate the direct differentiation to the β-cell phenotype. The α-cells, β-cells and δ-cells secrete glucagon, insulin and somatostatin, respectively. The pancreatic polypeptide cells secrete pancreatic polypeptide which regulates the secretory activity of the pancreas.

**Table 1 cells-11-02145-t001:** Advantages and disadvantages of the use of viral vectors in gene therapy and their delivery of β-cell transcription factors to generate insulin-producing cells.

Vector	Advantages	Disadvantages	Delivery of Transcription Factors	References
**Adenoviral**	Transfers genes to dividing and non-dividing cells Large packaging capacity (~8 kB) Produce high virus titres Gene delivery at high multiplicity of infection	Transient expression Highly immunogenic	*Pdx1* only *Pdx1, Ngn3* and *MafA* *Pdx1, NeuroD1* and *MafA* *Pdx1, NeuroD1, MafA* and *Isl1*	[64,65,68,69,70,71,72,73,74,75,76,77,78]
**Adeno-Associated**	Transfers genes to dividing and non-dividing cells Integrates into host genome with stable gene expression Low immunogenicity	Low packaging capacity (~5 kB) Neutralising antibodies can be generated Requires helper virus for effective use	*Pdx1* only	[75]
**Lentiviral**	Transfers genes to dividing and non-dividing cells Integrates into host genome with stable gene expression Large packaging capacity (~10 kB) Third-generation vectors are self-inactivating No known immunogenic proteins created Produce high virus titre (108 TU/mL)Simple system for vector manipulation and production	Possibility of insertional mutagenesis in host genome in second-generation LV Contains three HIV-1 genes (gag, pol, rev)	*Pdx1/Pdx1-VP16* only	[79,80]

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
