# Peer review of "Pancreatic Transdifferentiation Using β-Cell Transcription Factors for Type 1 Diabetes Treatment"

_cells, 2022, doi:10.3390/cells11142145_

Round 1

Reviewer 1 Report

This review outlines strategies and approaches for transdifferentiation of pancreatic beta cells and discusses this idea as a treatment for Type 1 Diabetes. The work is accessible to nonspecialists, has simple, easy to read figures/tables and makes good use of existing literature from related fields of stem cell biology and beta cell differentiation. A few minor suggestions are noted:

1. Although some of the difficulties with achieving transdifferentiation are pointed out, it should also be noted that 'transdifferentiation' as a term in the field does not have a strong consensus and is still quite controversial. The actual phenotype of the transdifferentiated beta-like cells is far from completely resembling true beta cells and this is a major roadblock to this sort of approach for treatment- e.g. how do you know when you have 'fully transdifferentiated' beta-like cells from another lineage? This should be at least acknowledged in the manuscript.

2.  Some citations around recent clinical trials on iPSC-derived beta cells for T1D are missing. Authors should at least mention the advances made by the work of Ramzy et al. Cell Stem Cell 2021 and Shapiro et al. 2021 Cell Reports Medicine which observed stimulated C-peptide release in iPSC-beta cell encapsulated devices, despite lack of long-term and stable insulin secretion.

3. In the 'Transdifferentiation of Stem cells' section, I believe is a misnomer to refer to embryonic/pluripotent stem cells as transdifferentiating. To be pluripotent means cells can be differentiated along any of the three germ layer lineages (ecto, endo, mesoderm). Therefore, this should be 'differentiation' not transdifferentiation. Also statements in lines 163-164 should be corrected in this regard.

Author Response

  1. Although some of the difficulties with achieving transdifferentiation are pointed out, it should also be noted that 'transdifferentiation' as a term in the field does not have a strong consensus and is still quite controversial. The actual phenotype of the transdifferentiated beta-like cells is far from completely resembling true beta cells and this is a major roadblock to this sort of approach for treatment- e.g. how do you know when you have 'fully transdifferentiated' beta-like cells from another lineage? This should be at least acknowledged in the manuscript.

I have added in the criteria to decipher when exactly the transdifferentiation process has occurred at line 146-150. I have specified the components of transdifferentiated β-like cell on line 157-159.

  1. Some citations around recent clinical trials on iPSC-derived beta cells for T1D are missing. Authors should at least mention the advances made by the work of Ramzy et al. Cell Stem Cell 2021 and Shapiro et al. 2021 Cell Reports Medicine which observed stimulated C-peptide release in iPSC-beta cell encapsulated devices, despite lack of long-term and stable insulin secretion.

I have read these papers and they reference embryonic stem cells rather than iPSCs? I have included a brief description of these papers at line 171-175.

  1. In the 'Transdifferentiation of Stem cells' section, I believe is a misnomer to refer to embryonic/pluripotent stem cells as transdifferentiating. To be pluripotent means cells can be differentiated along any of the three germ layer lineages (ecto, endo, mesoderm). Therefore, this should be 'differentiation' not transdifferentiation. Also, statements in lines 163-164 should be corrected in this regard.

This wording has been changed at line 163 and 171.

Reviewer 2 Report

In this review, Mahoney et al summarise the state of the art in an increasingly important field of research relating to beta-cell regeneration via transduction with viral vectors encoding relevant transcription factors. They summarise the area well and provide a succinct and readily comprehensible analysis that will be of value to the field. A couple of issues might be considered further:

1. The concept of recurrent autoimmunity is raised as a potential issue when using patient cells as the vehicles to regenerate beta-cells but this is not considered in detail. In particular, it would be helpful to gain a perspective from the authors on the likely barriers to successful mitigation. They argue that transducer beta-cells may not express a normal complement of islet antigens - is there evidence to support this interesting idea? 

2. Figure 1 is helpful but it is unclear whether it refers mainly to mice or to human development. It would be helpful to indicate where and how these differ during pancreatic development.

3. The transdifferentiation of liver cells is used as an example of beta-cell regeneration and, while the authors emphasise the acquisition of beta-cell features, they do not highlight whether the phenotype of the cells is such that liver proteins are lost. A more complete description of these outcomes (and an indication of what exactly is required to comprise a functionally competent beta-cell) would be helpful to the reader.

Author Response

  1. The concept of recurrent autoimmunity is raised as a potential issue when using patient cells as the vehicles to regenerate beta-cells but this is not considered in detail. In particular, it would be helpful to gain a perspective from the authors on the likely barriers to successful mitigation. They argue that transducer beta-cells may not express a normal complement of islet antigens - is there evidence to support this interesting idea? 

I have included an explanation at line 212-218 that autoimmune destruction was not observed in insulin secreting liver cells or pituitary cells.

  1. Figure 1 is helpful but it is unclear whether it refers mainly to mice or to human development. It would be helpful to indicate where and how these differ during pancreatic development.

The figure is based on transcription factors involved in human pancreas development, this is now specified in the caption of Figure 1.

  1. The transdifferentiation of liver cells is used as an example of beta-cell regeneration and, while the authors emphasise the acquisition of beta-cell features, they do not highlight whether the phenotype of the cells is such that liver proteins are lost. A more complete description of these outcomes (and an indication of what exactly is required to comprise a functionally competent beta-cell) would be helpful to the reader.

At line 240-249 I have addressed this comment and said that liver cells are able to maintain their function while being able to store and secrete insulin. I have also added in what a transdifferentiated cell would need to do in order to be considered a functional β-cell at line at 157-159.

Reviewer 3 Report

This review article is nicely written in wide and fair discussion basis.

Please check English, sentences carefully.

Example: Line 296, vacant, appropriate word must be missed, need to be added.

Author Response

 Line 296, vacant, appropriate word must be missed, need to be added.

The sentence has been changed at line 323.